# Mitochondrial ROS–ER Stress Axis Governs IL-10 Production in Neutrophils and Regulates Inflammation in Murine *Chlamydia pneumoniae* Lung Infection

**DOI:** 10.3390/cells14191523

**Published:** 2025-09-29

**Authors:** Bin Chou, Kazunari Ishii, Yusuke Kurihara, Akinori Shimizu, Michinobu Yoshimura, Ryo Ozuru, Ryota Itoh, Atsuhiko Sakamoto, Kenji Hiromatsu

**Affiliations:** 1Department of Microbiology & Immunology, Faculty of Medicine, Fukuoka University, Fukuoka 814-0180, Japan; choubin@fukuoka-u.ac.jp (B.C.); kurihara_yusuke@kurume-u.ac.jp (Y.K.); shimizua@fukuoka-u.ac.jp (A.S.); myoshimura@fukuoka-u.ac.jp (M.Y.); ozuru@fukuoka-u.ac.jp (R.O.); ryito@fukuoka-u.ac.jp (R.I.); a.sakamoto.sw@fukuoka-u.ac.jp (A.S.); 2Department of Infectious Medicine Division of Eukaryotic Microbiology, Faculty of Medicine, Kurume University, Fukuoka 830-0011, Japan

**Keywords:** IL-10, neutrophil, *Chlamydia pneumoniae*, ER stress, IRE1α

## Abstract

Neutrophils are among the first cells to be recruited to the lungs during *Chlamydia pneumoniae* infection in mouse models; however, their regulatory functions are not yet fully understood. This study examined the mechanisms and significance of IL-10-producing neutrophils throughout *C. pneumoniae* pulmonary infection in C57BL/6 mice. Our findings revealed that infection with *C. pneumoniae* induces IL-10 secretion in bone marrow-derived neutrophils, depending on Toll-like receptor 2 (TLR2) activation. This process involves TLR2-dependent mitochondrial reactive oxygen species (ROS) production, which triggers the endoplasmic reticulum (ER) stress pathway, including IRE1α and subsequent Xbp1 splicing. Inhibition of this pathway or depletion of neutrophils (using the 1A8 monoclonal antibody) significantly reduces IL-10 levels in bronchoalveolar lavage fluid (BALF) in vivo. Conversely, the absence of IL-10-producing neutrophils, whether through depletion or TLR2 deficiency, leads to increased IL-12p70 and IFN-γ-positive NK cells, along with decreased regulatory T cells and M2-like macrophages. This results in a lower bacterial burden in the lungs but causes more severe pulmonary damage and decreased survival rates. These findings highlight that IL-10 produced by neutrophils via the TLR2-mitochondrial ROS–ER stress pathway is essential for modulating pulmonary immune responses and maintaining immune homeostasis during *C. pneumoniae* infection, thereby preventing excessive inflammation and tissue damage.

## 1. Introduction

*Chlamydia pneumoniae* is a widespread intracellular bacterium responsible for both acute and chronic respiratory infections, with a significant clinical burden worldwide [1,2]. Although host innate immunity plays a pivotal role in controlling bacterial proliferation, dysregulated inflammation contributes to lung pathology and worsened outcomes [3,4]. Neutrophils constitute the earliest and most abundant innate immune cells recruited to infected lungs, traditionally regarded as antimicrobial effector cells that mediate tissue damage through robust proinflammatory responses [5,6,7,8]. However, recent evidence suggests neutrophils display remarkable plasticity, producing diverse cytokines including anti-inflammatory mediators such as interleukin-10 (IL-10), which critically modulate immune responses and tissue homeostasis [9,10,11,12,13]. Despite growing recognition of IL-10’s role in shaping immunity in various infectious diseases, the mechanisms controlling IL-10 production specifically by neutrophils during bacterial pneumonia remain poorly understood.

The endoplasmic reticulum (ER) is a vital organelle in eukaryotic cells that handles protein synthesis, folding, modification, and transport, along with lipid production and calcium storage [14]. ER stress is a condition marked by the buildup of unfolded proteins in the ER, typically induced by factors such as hypoxia, calcium perturbation, or reactive oxygen species (ROS) [15]. Induction of endoplasmic reticulum (ER) stress impacts ER functions and triggers the unfolded protein response (UPR) to mitigate the effects of ER stress and restore ER functionality. In mammalian cells, UPR signaling is mediated through three distinct pathways, each regulated by an ER stress sensor: inositol-requiring enzyme 1α (IRE1α), protein kinase RNA-like endoplasmic reticulum kinase (PERK), and activating transcription factor 6 (ATF6) [16]. ER stress will be induced in immune cells under conditions such as bacterial infections, tumor microenvironments, autoimmune diseases, and other related circumstances. This stress will subsequently regulate the activation or polarization of immune cells [17,18,19,20]. Furthermore, ER stress can also be induced in neutrophils within the context of cancer, autoimmune diseases, and other conditions, thereby regulating the immune responses of these neutrophils [21,22,23]. Although those reports have elucidated the relationship between ER stress and immune responses, the mechanisms underlying the induction and functions of ER stress in neutrophils during bacterial infection remain unclear.

Toll-like receptor 2 (TLR2) is a key pattern recognition receptor for *C. pneumoniae* components, mediating downstream signaling that influences inflammation and host defense [24,25]. Separately, mitochondrial ROS (mtROS) and ER stress pathways have emerged as essential regulators of innate immune signaling and cytokine production in multiple cell types [17,26,27,28]. Nevertheless, their interplay governing neutrophil IL-10 secretion in the context of pulmonary chlamydial infection has not been elucidated.

In this study, using murine models of *C. pneumoniae* lung infection and comprehensive in vitro approaches, we investigate the cellular sources, upstream signaling pathways, and immunological functions of neutrophil-derived IL-10. We reveal a novel TLR2-dependent mtROS–ER stress–IRE1α axis as a critical driver of neutrophil IL-10 production, balancing host antimicrobial defense and immunopathology. These findings provide unprecedented insight into innate immune regulation and identify potential targets for therapeutic modulation in respiratory bacterial infections.

## 2. Materials and Methods

### 2.1. Mice

C57BL/6 (WT) mice were purchased from Kyudo (Tosu, Japan). C57BL/6J-background TLR2^−/−^ mice were purchased from Oriental BioService (Kyoto, Japan). All mice were kept in a specific pathogen-free facility. The animal experiments were carried out in accordance with protocols approved by the Animal Care and Use Committee of Fukuoka University. All of the mice used in this study were female and 8 to 10 weeks old, and a 25% body weight loss was established as a humane euthanasia endpoint for all in vivo mouse experiments.

### 2.2. Chlamydia and Infection

*C. pneumoniae* (strain: AR-39, ATCC 53592, and strain: CWL-029, ATCC VR-1310) and *Chlamydia muridarum mouse pneumonitis* (MoPn) (strain: Nigg II, ATCC VR-123) were purchased from the American Type Culture Collection. *C. pneumoniae* and MoPn were prepared in HEp-2 cells and propagated using a previously reported method [29]. Purification of EBs was carried out as described earlier [30], with the Renografin gradient centrifugation repeated three times to ensure the purity of the EB preparations. The purified EBs were suspended in sucrose/phosphate/glutamate (SPG) medium (75.0 g sucrose, 0.52 g KH_2_PO_4_, 3.07 g Na_2_HPO_4_·12H_2_O, 0.72 g glutamic acid; distilled water to 1000 mL) and stored at –80 °C until further use. All *Chlamydia* stocks were confirmed negative for *Mycoplasma* contamination using a MycoAlert Mycoplasma detection kit (Lonza, Basel, Switzerland). Serial 10-fold dilutions of the purified EBs suspension were added to HEp-2 cells and, 30 h later, stained with a FITC-conjugated anti-Chlamydia lipopolysaccharide antigen monoclonal antibody (PROGEN, Heidelberg, Germany). This process was used to count the infectious EBs, which are defined as inclusion-forming units (IFU)/mL.

Mice were anesthetized and given an intranasal inoculation with 1 × 10^5^ IFU of *C. pneumoniae* in a 50 μL final volume of SPG. Their body weight was monitored daily after infection. BALF was obtained from mice by inserting an intravenous catheter into the trachea and washing the lungs twice with 0.6 mL of PBS. Mice were euthanized at different days post-infection, and the lungs were collected aseptically. The chlamydial growth in the lung was determined by the IFU assay.

### 2.3. Depletion of Neutrophils In Vivo

Mice were intraperitoneally (i.p.) injected with 200 μg/mouse anti-Ly-6G antibody (clone: 1A8, Bio X cell, Lebanon, NH, USA) one day before and three days after the infection with chlamydia. Mice in the control group were i.p. injected with 200 μg per mouse of Rat IgG2a monoclonal antibody (clone: RTK2758, BioLegend, San Diego, CA, USA) as the isotype control antibody. Anti-Gr-1 antibody (anti-Ly-6G/Ly-6C, Clone: RB6-8C5) was purified from the supernatant of a hybridoma. For in vivo depletion, mice were i.p. injected with 200 μg anti-Gr-1 antibody per mouse on the same schedule as that of the anti-Ly-6G antibody.

### 2.4. Cells and Reagents

Bone marrow-derived PMNs were isolated from the femurs of WT mice using a neutrophil isolation kit (Miltenyi Biotec, Bergisch Gladbach, Germany). Neutrophils were suspended in a growth medium (RPMI 1640, 10% heat-inactivated FBS, 10 mM HEPES, and 100 μg/mL streptomycin sulfate). TLR2 inhibitor C29 and IRE1α inhibitor 4μ8C (Selleck Chemicals, Houston, TX, USA) were dissolved in DMSO at a concentration of 20 mM and stocked at −80 °C. The final concentration of 4μ8C was 25 μM to be added to the culture medium of neutrophils. TLR4 inhibitor TAK-242 and TLR9 inhibitor E6446 were purchased from Selleck Chemicals. Mitochondrial ROS scavenger MitoTEMPO (Sigma-Aldrich Co., St. Louis, MO, USA) was dissolved in DMSO at a concentration of 10 mM and stored at −20 °C. Alexa flour 647 anti-XBP-1s antibody (Q3-695, BD Biosciences, Franklin Lakes, NJ, USA) was used to stain XBP-1 protein in neutrophils.

### 2.5. In Vitro and In Vivo Treatment with 4-PBA

The utilized 4-Phenylbutyric acid (4-PBA) (Sigma-Aldrich Co.) was stocked at a concentration of 1 M. Bone marrow-derived PMN were isolated from WT mice and then plated into 48-well plates at a density of 6 × 10^5^ cells/well. PMN were infected at the multiplicity of infection (MOI) of 1. *C. pneumoniae* 4-PBA was added to the culture medium at the final concentration of 1 mM or 2 mM. The culture medium was collected 24 h after the infection. C57BL/6J mice were intranasally infected with *C. pneumoniae* and orally treated with 4-PBA at 120 mg/kg daily. Mice were sacrificed 4 days and 8 days after infection.

### 2.6. Cytokine ELISA

ELISA assays for IL-10, IL-12p70, IL-6, and TNF*α* were performed using the DuoSet ELISA development system (R&D Systems, Inc., Minneapolis, MN, USA) according to the manufacturer’s instructions. ELISA plates were read via a Model 680 microplate reader (BIO-RAD Laboratories, Inc., Hercules, CA, USA).

### 2.7. LDH Cell Cytotoxicity Assay

LDH (lactate dehydrogenase) release from dead cells was measured using the CytoTox 96 Non-Radioactive Cytotoxicity Assay Kit (Promega, Madison, WI, USA) as previously described [31]. Briefly, cells were lysed with 0.8% Triton X-100 (Tx-100), a non-ionic detergent, for 45 min to release intracellular LDH. This method was also employed to determine the maximum LDH release as a reference in cell cytotoxicity assays.

### 2.8. Flow Cytometric Determination of Mitochondrial ROS, Mitochondrial Membrane Potential, and XBP1s Protein in Neutrophils

Mitochondrial ROS levels were measured using MitoSOX (Thermo Fisher Scientific, Waltham, MA, USA) staining. Bone marrow-derived neutrophils were incubated with the mitochondrial superoxide-specific dye MitoSOX (1 μM) at 37 °C for 20 min to detect mitochondrial ROS 6 h after mock, *C. pneumoniae*, or *C. muridarum* infection. For the measurement of mitochondrial membrane potential, infected neutrophils were stained with MitoTracker Green (250 nM, Thermo Fisher Scientific) and MitoTracker Red (500 nM, Thermo Fisher Scientific), and live cells were identified using a Zombie NIR™ Fixable Viability Kit (BioLegend) and detected by flow cytometry (FACSCanto II, BD). Data analyses and plotting were performed using FlowJo software, version 10.8.1 (Tree Star, Inc., Ashland, OR, USA).

### 2.9. Isolation of Mononuclear Cells from the Lung Tissues of Mice

Mice lungs were surgically excised and subsequently minced into approximately 1–2 mm fragments. Following incubation with Type I collagenase (SCR103, Sigma-Aldrich Co.) at 37 °C for 30 min, the lung tissue was homogenized, and the total lung cells were isolated by centrifugation at 500× *g* for 10 min. Lung lymphocytes were subsequently separated from the total lung cell population using Percoll (Cytiva, Marlborough, MA, USA). Briefly, 5 mL of 65% Percoll was added to a 15 mL tube, and the total lung cells suspended in 5 mL of 45% Percoll were carefully overlayed onto the 65% Percoll layer. After centrifugation at 1000× *g* for 22 min, the lung mononuclear cells were collected from the interface between the 45% and 65% Percoll layers and utilized for further flow cytometry analysis.

### 2.10. Flow Cytometric Analysis and Intracellular Cytokine

Flow cytometric analysis using antibodies of anti-CD4, anti-Gr-1, anti-CD11b, anti-Ly-6G, anti-NK1.1, anti-IFNγ, and anti-Foxp3 were purchased from BioLegend. For flow cytometry, cells were isolated from the lungs of both control and neutrophil-depleted mice. Cells were stained with antibodies and analyzed using FACS Canto II. For flow cytometric analysis of intracellular cytokine, lymphocytes were isolated from the lung of control mice or neutrophil-depleted mice and cultured (1 × 10^6^/mL) with 10 ng/mL of PMA (Sigma-Aldrich Co.) plus 500 ng/mL of Ca^2+^ ionophore (Sigma-Aldrich CO.) in the presence of GolgiStop plus GolgiPlug (BD Biosciences). Cells were then stained with anti-NK1.1, anti-CD4, fixed, and permeabilized with the Foxp3 staining buffer kit (eBiosciences, San Diego, CA, USA) according to the manufacturer’s directions. Cells were then stained with anti-IFN-γ or anti-Foxp3 and analyzed for the expression of cytokine using FACS Canto II.

### 2.11. Immunofluorescence

*Chlamydia*-infected neutrophils were fixed with 4% paraformaldehyde, blocked, and permeabilized with 1% BSA and 0.1% Triton X-100 in PBS for 5 min at RT. The cells were incubated with FITC-conjugated primary antibodies against chlamydial multi-epitope containing Evans blue as a counterstain (PROGEN) for 2 h at RT. After washing with PBS, stained coverslips were mounted with ProLong^TM^ Glass Antifade Mountant with NucBlue^TM^ (Thermo Fisher Scientific). Images were obtained using confocal microscopy, LSM 710 (Carl Zeiss AG, Oberkochen, Germany).

### 2.12. Immunoblot Analysis

The cells were washed in PBS and lysed with RIPA buffer (containing 50 mM Tris [pH 7.4], 1% Nonidet P-40, 0.5% sodium deoxycholate, 0.1% SDS, 150 mM NaCl, 2 mM EDTA, and 50 mM NaF), supplemented with a protease inhibitor cocktail (NACALAI TESQUE, INC., Kyoto, Japan) and PhosSTOP phosphatase inhibitor cocktail (Sigma-Aldrich Co.). After lysis, the cells were briefly sonicated. Protein concentrations were determined using a BCA (bicinchoninic acid) assay kit (Thermo Fisher Scientific), and equal protein amounts were loaded onto SDS-PAGE gels. Proteins were transferred to PVDF membranes (BIO-RAD Laboratories, Inc.) with a Trans-Blot SD semi-dry transfer cell (BIO-RAD Laboratories, Inc.) following the manufacturer’s instructions. Membranes were blocked with Blocking One (NACALAI TESQUE, INC.) for 30 min. Subsequently, they were incubated overnight at 4 °C with primary antibodies diluted in Can Get Signal solution 1 (Toyobo Co., Ltd., Osaka, Japan). The following day, membranes were incubated for 1 h at room temperature with HRP-conjugated secondary antibodies targeting rabbit or mouse IgG (Cell Signaling Technology, Inc., Danvers, MA, USA), diluted in Can Get Signal solution 2 (Toyobo Co., Ltd.). Immunoreactive bands were visualized using ECL blotting reagents EzWestLumi plus (ATTO CORPORATION, Tokyo, Japan) and detected with LAS-3000 (FUJIFILM, Tokyo, Japan). Densitometric analysis was performed using ImageJ software, version 1.53q.

### 2.13. Histology and Immunochemistry

Mouse lung tissues fixed in formalin and embedded in paraffin were sliced into 4 µm sections and stained with eosin and hematoxylin. Deparaffinized sections were treated with 3% H_2_O_2_ in methanol to inactivate endogenous peroxidase after pH 6.0 retrieval for immunochemistry staining. After incubation with the anti-IL-10 antibody (BS-0698R, Bioss Inc., Woburn, MA, USA), the sections were stained with Histofine Simple Stain MAX-PO (414341, NICHIREI BIOSCIENCES INC., Tokyo, Japan) and N-Histofine^®^ DAB substrate Kit (425011, NICHIREI BIOSCIENCES INC.). Subsequently, the specimens were incubated with a solution of 0.1 mg/mL naphthol AS-D chloroacetate (11591, MUTO PURE CHEMICALS CO., LTD., Tokyo, Japan) and 0.1 mg/mL Fast-Garnet GBC (09235-54, NACALAI TESQUE, INC.) at 37 °C for 30 min. Finally, they were counterstained with Hematoxylin (3002-2, MUTO PURE CHEMICALS CO., LTD.). Histopathological scores of lung tissue were evaluated based on the parameters in Appendix A [32,33].

### 2.14. Quantitative RT-PCR

Total mRNAs were isolated from neutrophils by using the ISOGEN II agent (NIPPON GENE CO., LTD., Tokyo, Japan). cDNA libraries were reverse transcribed from total mRNA by using PrimeScript RT reagent Kit with gDNA Eraser (Takara Bio Inc., Kusatsu, Japan). Real-time PCRs were performed by using SYBR Green Realtime PCR Master Mix (Toyobo Co., Ltd.) and on the 7500 real-time PCR system (Thermo Fisher Scientific). All primer sequences used in the current study are listed in Appendix A. The thermal cycling conditions included 30 s at 95 °C, followed by 40 cycles of denaturation at 95 °C for 5 s, annealing at 55 °C for 30 s, and extension at 72 °C for 34 s.

### 2.15. Xbp1 Splicing Assay

Activation of XBP1 can be measured by limited enzyme digestion of m*Xbp1* PCR production [23]. Briefly, cDNA of PMN total RNA was isolated as described above. Xbp1 transcripts were amplified by using XBP1 primer set listed in Appendix A. The PCR products were purified and digested by PstI at 37 °C 2 h. Following these steps, un-spliced *Xbp1* (us *Xbp1*) was cut to two fragments 254 bp and 215 bp, while spliced *Xbp1* (s*Xbp1*) was not cut, totalling a size of 469 bp. The PstI-digested DNA fragments were loaded onto a 2% agarose gel to distinguish between us *Xbp1* and s *Xbp1*. ImageJ software was used to analyze band density, and the relative ratio of s *Xbp1*/total *Xbp1* was compared with control treatment PMN infected with *C. pneumoniae*.

### 2.16. Statistical Analysis

Data are expressed as means ± SD. Depending on the experiment, differences between groups were analyzed using unpaired two-tailed Student’s *t*-test, a one-way ANOVA followed by Tukey’s multiple comparison test, or a two-way ANOVA followed by Tukey’s multiple comparison test. Survival curves were created with the Kaplan–Meier method and compared via the log-rank test. All analyses were conducted using GraphPad Prism software, version 9.5.1 (GraphPad, La Jolla, CA, USA). * *p* < 0.05, ** *p* < 0.01, *** *p* < 0.001, and **** *p* < 0.0001 indicate statistically significant results.

## 3. Results

### 3.1. C. pneumoniae Live Infection in Mouse Neutrophils Triggers Robust IL-10 Production via TLR2

C57BL/6 bone marrow (B/M)-derived neutrophils purified by MACS (purity over 98%, Appendix A) were infected with *C. pneumoniae* in vitro and examined for mRNA induction of various cytokines 24 h post-infection. It was observed that in addition to mRNA for proinflammatory cytokines (IL-1β, IL-6, TNFα), there was a significant increase in IL-10 mRNA in B/M neutrophils 24 h after infection with *C. pneumoniae* (Figure 1A). The MOI-dependent secretion of IL-10 protein from neutrophils at 24 h post-infection was confirmed by ELISA (Figure 1B). This induction of IL-10 in neutrophils occurred not only through infection with *C. pneumoniae* strain AR-39 but also with strain CWL-029, and it depended on live *C. pneumoniae* but not heat-killed *C. pneumoniae* (Figure 1C). *C. pneumoniae* infection-induced IL-10 production in neutrophils was significantly suppressed by the TLR2 inhibitor C29 but not by inhibitors targeting TLR4 or TLR9 (Figure 1D). There was no significant difference in the induction of TNF-α or cell death in *C. pneumoniae*-infected neutrophils treated with or without the TLR2 inhibitor C29 (Appendix A). Furthermore, similar levels of *C. pneumoniae* infectious burden in neutrophils treated with or without C29 were confirmed by immunofluorescence staining and quantitative PCR detecting 16S rDNA of *C. pneumoniae* (Figure 1E,F), indicating that treatment with the TLR2 inhibitor C29 did not negatively affect the efficiency of *C. pneumoniae* entry and proliferation in neutrophils. Finally, TLR2-dependent IL-10 production in *C. pneumoniae*-infected neutrophils was confirmed using B/M neutrophils from TLR2^−/−^ mice (Figure 1G). These results show that *C. pneumoniae* infection triggers robust IL-10 secretion in mouse neutrophils, which is blocked explicitly by TLR2 inhibition without affecting bacterial uptake or survival, highlighting a targeted regulatory pathway.

### 3.2. Mitochondrial ROS and ER Stress–IRE1α Axis Are Essential for Neutrophil IL-10 Production After C. pneumoniae Infection

Next, we investigated the underlying mechanisms of *C. pneumoniae* infection-induced IL-10 production in neutrophils. It has been reported that ER stress can be induced in neutrophils [23], and the induction of IL-10 can also be regulated by ER stress [20]. We previously reported that *C. pneumoniae* infection induces ER stress and the unfolded protein response (UPR), leading to increased levels of mitochondrial ROS (mtROS) in murine adipocytes [34]. ER stress can be triggered by mtROS in immune cells and influences the induction of certain cytokines [35,36]. Therefore, we examined whether mtROS and ER stress are connected to *C. pneumoniae* infection-induced IL-10 production from neutrophils. The gate comprising cells stained with the mitochondrial mass indicator MitoGreen at high levels and the mitochondrial membrane potential indicator MitoRed at low levels signifies the proportion of dysfunctional mitochondria, which are characterized by a loss of membrane integrity potential [31,37]. We found that the proportion of MitoGreen ^high^ and MitoRed ^low^ increased at 6 h post-infection compared to mock infection in neutrophils from WT mice but not from TLR2^−/−^ mice (Figure 2A). Mitochondrial ROS in WT neutrophils increased after infection with *C. pneumoniae* compared to mock infection, but this increase was not seen in TLR2^−/−^ neutrophils (Figure 2B). Furthermore, RT-PCR analysis of ER stress-related gene expressions in neutrophils at 12 h after infection with *C. pneumoniae* revealed that the mRNA expression of *Hspa5* (also known as *Grp78/Bip*) encoding protein folding chaperone Bip and *IRE1a* genes encoding the protein inositol-requiring enzyme 1α (IRE1α) (which is a crucial component of the UPR) was upregulated in neutrophils after infection with *C. pneumoniae* (Figure 2C). The increase in Bip and IRE1α expression at protein levels was also confirmed by Western blotting of neutrophils at 12 h and 24 h after infection with *C. pneumoniae* (Figure 2D). Furthermore, to investigate the relationship between mtROS and ER stress, the production of mtROS in *C. pneumoniae*-infected neutrophils co-cultured with or without the ER stress inhibitor 4-phenylbutyric acid (4-PBA) was measured using flow cytometry. Rotenone, an mtROS inducer, served as a positive control. The infection-induced increase in mtROS in neutrophils was not prevented by treatment with the 4-PBA (Figure 2E,F), indicating that the increase in mtROS precedes ER stress in neutrophils subsequent to *C. pneumoniae* infection. Treatment with MitoTEMPO, an mtROS scavenger, inhibited the increase in IRE1α and Bip expression in infected neutrophils (Figure 2G). *Xbp1* mRNA is known as the downstream substrate of IRE1α endoribonuclease [38]. When IRE1α is activated, un-spliced *Xbp1* mRNA is spliced out in a 23 bp fragment and produced as the spliced form of *Xbp1* to be translated into XBP1 protein [23]. Due to the cutting site of restriction enzyme *Pst* I included in that 23 bp fragment, PCR products of un-spliced *Xbp1* (us*Xbp1*) and spliced *Xbp1* (s*Xbp1*) can be distinguished by cutting with PstI. Therefore, to confirm the activation of IRE1α in neutrophils after infection with *C. pneumoniae*, MitoTEMPO and 4μ8C, an inhibitor of IRE1α, were added to the culture medium of neutrophils infected with *C. pneumoniae*. Interestingly, compared to mock-infected neutrophils, s*Xbp1* was significantly upregulated in neutrophils after infection with *C. pneumoniae*. The induction of s*Xbp1* in infected neutrophils was notably reduced by co-culture with 4μ8C and MitoTEMPO (Figure 2H). Using sXBP1-specific antibody and flow cytometry, we confirmed that the spliced form of XBP-1 was induced in *C. pneumoniae*-infected neutrophils, which was suppressed by adding MitoTEMPO or 4μ8C (Figure 2I,J). Next, to investigate the relationship between ER stress activation and IL-10 induction in neutrophils following *C. pneumoniae* infection, mtROS scavenger MitoTEMPO, ER stress inhibitor 4-PBA or IRE1α inhibitor 4μ8C were added to the culture medium of neutrophils after infection. We found that the induction of IL-10 in *C. pneumoniae*-infected neutrophils was inhibited by culturing with MitoTEMPO, but TNFα production and cell death in *C. pneumoniae*-infected neutrophils were unaffected by the addition of MitoTEMPO (Figure 2K and Appendix A). Induction of IL-10 in *C. pneumoniae*-infected neutrophils was also strongly inhibited by co-culturing with 4-PBA or 4μ8C, but cell death of *C. pneumoniae*-infected neutrophils was not affected by adding 4-PBA or 4μ8C (Figure 2L,M and Appendix A). These results establish the critical role of the mtROS–ER stress–IRE1α-XBP1 axis in neutrophil IL-10 induction.

### 3.3. C. muridarum Infection Neither Increases mtROS, Provokes ER Stress, nor Causes Neutrophils to Produce IL-10

In contrast to *C. pneumoniae*, *C. muridarum* did not cause rapid mitochondrial disruption or increase mitochondrial ROS/ER stress marker genes or Bip and IRE1α protein levels (Appendix A). *C. muridarum* infection did not trigger IL-10 in neutrophils (Appendix A), despite maintaining inflammatory cytokine production. Furthermore, *C. muridarum* intranasal lung infection in WT mice did not significantly increase IL-10 production in BALF, nor was there any dependence on TLR2 or effects of neutrophil depletion on IL-10 production (Appendix A). This difference highlights the pathogen-specific nature of this immunomodulatory circuit.

### 3.4. In Vivo, Neutrophil-Derived IL-10 Shapes Immune Responses in the Lungs of Mice Infected with C. pneumoniae

To investigate the in vivo role of IL-10-producing neutrophils in mice infected with *C. pneumoniae*, we depleted neutrophils by treating the mice with anti-Ly6G (1A8) mAb (Appendix A). We found that infected C57BL/6 WT mice demonstrated pronounced IL-10 in bronchoalveolar lavage fluid (BALF) on day 3 post-infection, which was abolished by neutrophil depletion (1A8 anti-Ly6G mAb) or TLR2 deficiency (Figure 3A), which confirms the in vitro findings. Notably, the amount of IL-12p70 in BALF was significantly higher in neutrophil-depleted mice and in TLR2^−/−^ mice compared to isotype-abs-treated WT mice (Figure 3B), while the levels of the inflammatory cytokine IL-6 in BALF showed no significant differences among these groups of mice (Figure 3C).The increased production of IL-12p70 in neutrophil-depleted mice or TLR2^−/−^ mice following infection indicates that macrophages and/or dendritic cells (DCs) within the pulmonary environment may produce elevated levels of IL-12p70 when IL-10-producing neutrophils are depleted or absent in the context of TLR2 deficiency. To investigate this hypothesis, anti-Gr-1 antibodies, which are specific to both Ly-6G and Ly-6C, were employed to deplete Ly6C^+^ macrophages and Ly6C^+^ conventional DCs [39,40], in addition to Ly6G^+^ neutrophils of mice in vivo (Appendix A). IL-10 levels in the BALF on day 3 after *C. pneumoniae* lung infection were significantly reduced in both anti-Ly-6G-treated and anti-Gr-1-treated mice, with no difference between these two groups (Appendix A). This indicates that IL-10 in BALF is primarily generated by neutrophils. On the other hand, the amount of IL-12p70 in BALF was significantly higher in anti-monospecific Ly-6G abs-treated mice but significantly lower in anti-Ly-6G/Ly-6C (Gr-1) abs-treated mice after infection with *C. pneumoniae* AR39 (Appendix A). These data suggest that IL-12p70 is predominantly generated by macrophages and/or DCs within the pulmonary tissue, whereas IL-10 is produced by neutrophils in the lungs subsequent to *C. pneumoniae* infection. Furthermore, IL-10 secreted by neutrophils may inhibit the production of IL-12p70 by macrophages and DCs during the early stages of infection with *C. pneumoniae*.

To further investigate how neutrophil depletion impacts immune responses after lung infection with *C. pneumoniae*, leukocytes infiltrating the lung were isolated and analyzed using flow cytometry. Our findings indicate that the proportions of F4/80^+^CD206^+^ M2-like macrophages and CD4^+^Foxp3^+^ regulatory T cells (Tregs) increased on day 3 following infection (Figure 3D,E). In comparison to control-infected WT mice, the induction of M2-like macrophages and Tregs was markedly diminished in mice treated with 1A8 monoclonal antibodies and in TLR2^−/−^ mice subsequent to infection with *C. pneumoniae* (Figure 3D,E), which aligns with the observed decrease in IL-10 levels in these subjects. Additionally, it was observed that the proportion of NK1.1^+^IFNγ^+^ cells in the lungs of mice treated with 1A8 monoclonal antibodies or TLR2^−/−^ mice was higher than that in control mice on day 3 post-infection with *C. pneumoniae* (Figure 3F). The *C. pneumoniae* burden in the lung determined by IFU assay showed that 1A8 mAb-treated mice or TLR2^−/−^ mice had a lesser bacterial burden than control WT mice at day 3 after infection (Figure 3G). To test whether the decrease in *C. pneumoniae* burden was due to increased activation of NK cells, mice were treated with anti-NK1.1 mAb (clone PK136) to deplete NK cells before infection with *C. pneumoniae*. The chlamydia burden in the lungs of mice treated with PK136 mAb was significantly worse than in control mice (Figure 3H).

### 3.5. ER Stress Inhibition Recapitulates the Neutrophil IL-10 Depletion Phenotype

Next, we examined whether systemic ER stress inhibition inhibits IL-10 production in the lung after infection with *C. pneumoniae*. As expected from the in vitro data shown in Figure 2L, the level of IL-10 in the BALF of *C. pneumoniae*-infected mice was significantly reduced by systemic treatment with the ER stress inhibitor 4-PBA (Figure 3I). Notably, IL-12p70 in BALF was inversely elevated in mice treated with 4-PBA after infection with *C. pneumoniae* (Figure 3J). Thus, these results suggest that systemic ER stress inhibition mirrored the immunophenotype of neutrophil depletion—impaired BALF IL-10, elevated IL-12p70, reduced regulatory cell frequencies, and worsened pathology—thus confirming the physiological relevance of the mitochondrial–ER stress axis in vivo.

### 3.6. In Vivo, Neutrophil-Derived IL-10 Mitigates Tissue Damage

Lastly, to further clarify the role of IL-10-producing neutrophils in vivo during *C. pneumoniae* lung infection in mice, immunohistopathological analysis of the lung was performed. We observed a significant increase in IL-10 expression in the lung after *C. pneumoniae* infection, which was suppressed in mice treated with 1A8 abs (Figure 4A). The proportion of IL-10-positive area was significantly lower in mice treated with 1A8 mAb compared to mice treated with isotype-matched control Abs (Figure 4B). The lung tissues at day 3 post-infection of mice treated with isotype-matched control mAbs revealed the presence of IL-10-positive neutrophils, which were stained with Chloroacetate esterase (Figure 4C). H/E staining of the lung tissues on days 3 and 7 after infection revealed that lung inflammation was significantly higher in 1A8 mAbs-treated mice compared to isotype Abs-treated mice (Figure 4D). The pathological score was significantly higher in 1A8 abs-treated mice at day 7 after infection with *C. pneumoniae* (Figure 4E). Although the lung bacterial burden at day 3 post-infection was reduced in 1A8-treated mice (Figure 3G), the survival rates in 1A8-treated mice were worse than in control isotype antibody-treated mice (Figure 4F). These results suggest that while early bacterial load was reduced with IL-10 loss, this immunological hyperactivation aggravated lung pathology, as evidenced by histological scoring, worse clinical outcome, and reduced survival.

## 4. Discussion

This study reveals an unexpected and novel immunoregulatory function of neutrophils as a major source of IL-10 during early pulmonary infection with *C. pneumoniae*. While neutrophils are conventionally regarded as pro-inflammatory effector cells mediating pathogen clearance, our findings demonstrate that these cells integrate TLR2 signals to trigger mitochondrial ROS production and ER stress sensor activation, culminating in robust IL-10 secretion. This mitochondrial ROS–ER stress–IRE1α–XBP1 axis represents a previously uncharacterized pathway controlling anti-inflammatory cytokine production specifically in neutrophils, differing from classical paradigms established in macrophages and T cells. The selective induction of IL-10 by *C. pneumoniae*—but not by closely related *C. muridarum*—underscores pathogen-specific immune modulation. Importantly, neutrophil-derived IL-10 balances antimicrobial defense and immunopathology, restraining excessive Th1-associated inflammation yet allowing controlled bacterial persistence. These insights reshape the current understanding of neutrophil plasticity and highlight the dynamic interplay between cellular metabolic stress pathways and innate immune regulation in bacterial pneumonia.

A key mechanistic insight is the identification of the TLR2–mitochondrial ROS–ER stress–IRE1α pathway as the central regulator of IL-10 production in *C. pneumoniae*-infected neutrophils. We observed that *C. pneumoniae* infection induced mitochondrial ROS, which subsequently activated ER stress, evidenced by upregulation of Hspa5 and IRE1α expression and splicing of *Xbp1* mRNA. The sequential nature of this pathway was confirmed by showing that mitochondrial ROS induction occurs upstream of ER stress and that inhibitors of mitochondrial ROS, ER stress, or IRE1α significantly suppressed IL-10 production. This specific pathway was not activated by *C. muridarum*, which also failed to induce IL-10 production in neutrophils, highlighting a pathogen-specific mechanism for immunomodulation.

The in vivo studies further elucidated the profound immunoregulatory functions of these IL-10-producing neutrophils. Depletion of neutrophils with 1A8 mAb or the absence of TLR2 led to a significant decrease in IL-10 levels in the BALF of infected mice. Concurrently, there was an inverse elevation of the Th1-inducing cytokine IL-12p70 in BALF from neutrophil-depleted or TLR2^−/−^ mice. This suggests that IL-10 released from neutrophils actively suppresses IL-12p70 production, likely by macrophages and/or dendritic cells, early in the infection. Furthermore, the lack of IL-10-producing neutrophils resulted in an increased proportion of IFN-γ-positive NK cells and a reduction in CD4^+^Foxp3^+^ regulatory T cells (Tregs) and F4/80^+^CD206^+^ M2-like macrophages in the lungs. These shifts indicate a promotion of Th1-type immunity and a suppression of regulatory immune responses when IL-10 from neutrophils is absent. Importantly, these changes in the immune landscape had a direct impact on the course of infection. Mice lacking IL-10-producing neutrophils (either through depletion or TLR2 deficiency) exhibited a reduced bacterial burden of *C. pneumoniae* in their lungs. This improved bacterial clearance is attributable to the enhanced Th1-type immunity, particularly the increased IFN-γ production by NK cells, which are known to be important for controlling chlamydial infection.

However, our findings reveal a crucial “double-edged sword” effect. Despite improved bacterial clearance, mice depleted of IL-10-producing neutrophils experienced more severe lung damage and worse survival rates than control mice. This indicates that although IL-10-producing neutrophils might not directly help eliminate bacteria, they are crucial for maintaining immune balance in the infected lung. By producing IL-10, these neutrophils regulate the immune response, preventing excessive inflammation and tissue injury, which can be harmful to the host, even if it allows some pathogen persistence. Therefore, IL-10-producing neutrophils are key to balancing effective antimicrobial immunity with the prevention of severe immunopathology during *C. pneumoniae* lung infection (Figure 5). Recently, N. Khan et al. reported that treating with β-glycan results in a distinct subset of immature neutrophils that depend on mitochondrial oxidative metabolism and produce IL-10, thereby enhancing disease tolerance to influenza A virus [41]. Our results add to the growing evidence about how neutrophils influence immunopathology and promote tissue repair. The present study provides important insights into how innate immune cells such as neutrophils actively influence the adaptive immune response and play a role in disease tolerance [42,43] within the context of intracellular bacterial infections.

## 5. Conclusions

In conclusion, our data identify neutrophils as critical modulators of pulmonary immune homeostasis during *C. pneumoniae* infection through a TLR2-dependent mitochondrial ROS and ER stress-mediated pathway that induces IL-10 production. This immunoregulatory mechanism attenuates detrimental hyperinflammation and tissue damage while delicately balancing pathogen control. Given the dual role of neutrophil-derived IL-10 in host defense and disease progression, targeting components of the mitochondrial stress response in neutrophils represents a promising but complex therapeutic avenue. Future studies defining how this axis is engaged across diverse infections and inflammatory settings will be essential for developing strategies to fine-tune immune responses and improve clinical outcomes in respiratory bacterial diseases.

## Figures and Tables

**Figure 1 cells-14-01523-f001:**
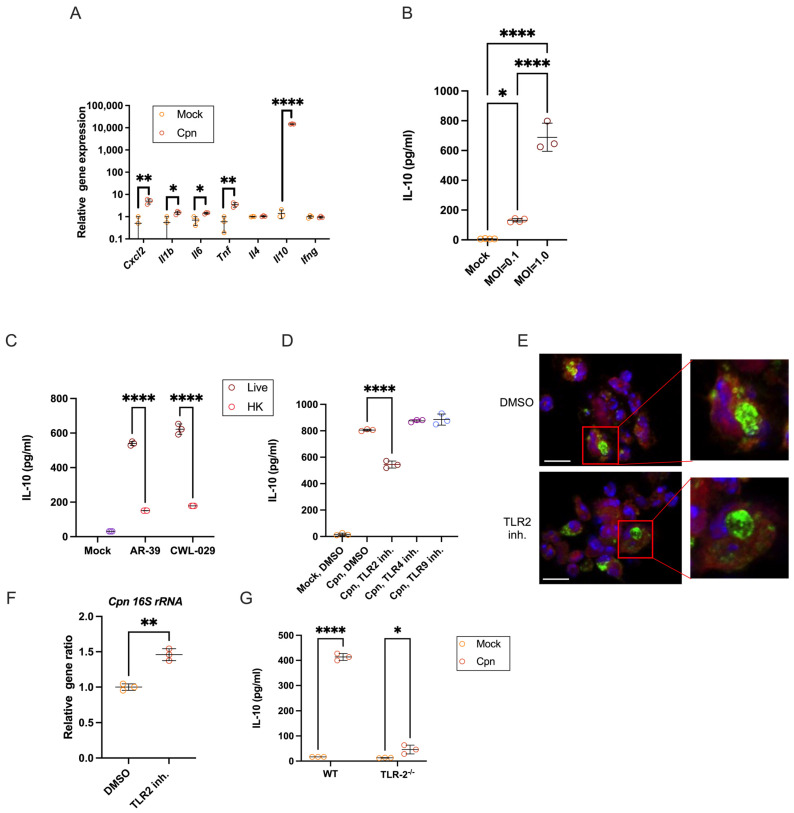
Live *C. pneumoniae* infection in mouse neutrophils elicits strong IL-10 production via TLR2. (**A**) Neutrophils were infected with *C. pneumoniae* (strain AR-39) at MOI = 1. mRNA was isolated from neutrophils after 24 h of infection. Expression of the indicated gene was confirmed by real-time PCR. (**B**) IL-10 levels in the supernatant were measured by ELISA. The supernatant from neutrophils infected with AR-39 at the specified MOI was collected after 24 h. (**C**) Neutrophils were infected with live or heat-killed *C. pneumoniae* strain AR-39 or strain CWL-029 at MOI = 1. The culture medium was collected 24 h post-infection. IL-10 in the medium was measured by ELISA. (**D**) Neutrophils from WT mice were infected with MOI = 1 AR-39 and cultured with TLR2 inhibitor C29 (100 μM), TLR4 inhibitor TAK-242 (1 μM), or TLR9 inhibitor E6446 (500 nM). Culture medium was collected 24 h after infection, and IL-10 levels were measured by ELISA. (**E**) Neutrophils from WT mice were infected with MOI = 1 AR-39 and co-cultured with or without TLR2 inhibitor C29 (100 μM) for 24 h, then stained with FITC-conjugated anti-chlamydia antibody. The bar length was 10 μm. (**F**) Neutrophils infected with MOI = 1 AR-39 were cultured with TLR2 inhibitor C29. After 24 h, total RNA was isolated, and *C. pneumoniae* 16S ribosomal RNA was quantified by real-time PCR. (**G**) Neutrophils from WT and TLR2^−/−^ mice were infected with MOI = 1 AR-39. Culture supernatants were collected after 24 h, and IL-10 levels were measured via ELISA. Statistical analyses were performed using an unpaired two-tailed Student’s *t*-test (**A**,**F**), a one-way ANOVA followed by Tukey’s multiple comparison test (**B**,**D**), and a two-way ANOVA followed by Tukey’s multiple comparison test (**C**,**G**). * *p* < 0.05, ** *p* < 0.01, and **** *p* < 0.0001, Each experiment was conducted at least three times with n ≥ 3, and the data presented are representative of these results. Cpn: *C. pneumoniae*.

**Figure 2 cells-14-01523-f002:**
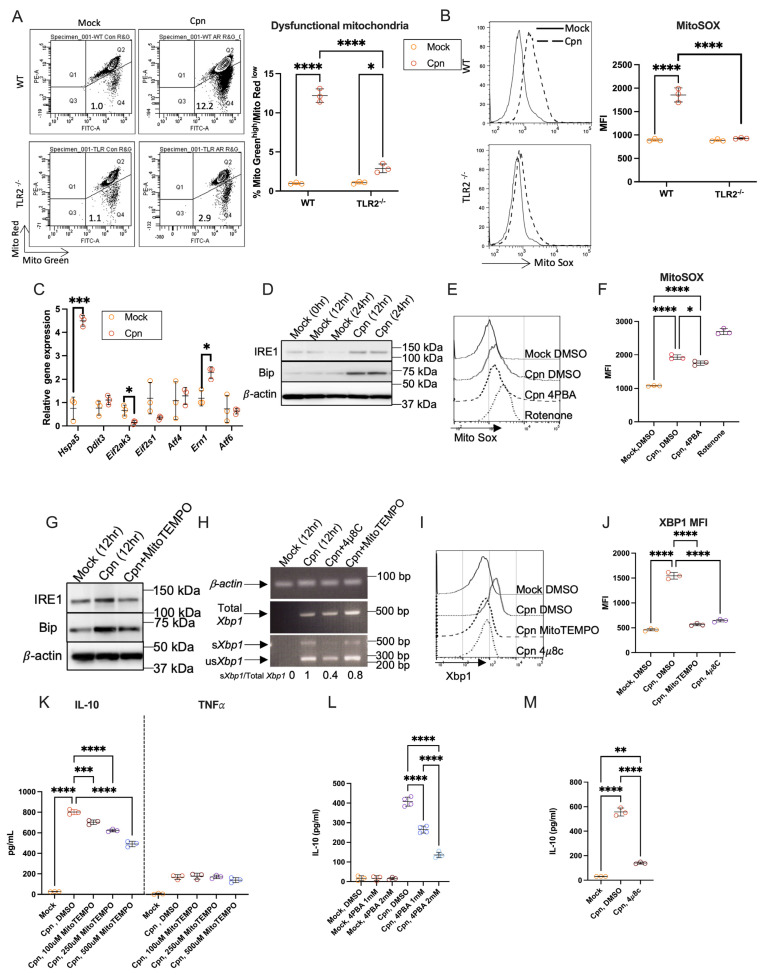
*C. pneumoniae* infection-induced mitochondrial ROS–ER stress–IRE1α sensor pathway activation leads to IL-10 production by murine neutrophils. B/M-derived neutrophils were infected with *C. pneumoniae* for 6 h, stained with MitoRed and MitoGreen (**A**) or with MitoSOX (**B**), and analyzed by flow cytometry. (**C**) ER stress-related genes in neutrophils at 12 h post-infection (h.p.i.) with mock or *C. pneumoniae* were analyzed by quantitative RT-PCR. (**D**) The protein of Bip or IRE1α, expressed in the mock or *C. pneumoniae*-infected neutrophils at an indicated time point, was assayed by Western blot. (**E**,**F**) Mock or *C. pneumoniae*-infected neutrophils were treated with the mtROS inducer Rotenone or the ER stress inhibitor 4-PBA. Neutrophils were collected at 12 h.p.i. and stained with MitoSOX, then analyzed by flow cytometry. (**G**) Mock or *C. pneumoniae*-infected neutrophils were treated with or without MitoTEMPO for 12 h. Bip and IRE1α proteins were detected by Western blot. (**H**) Mock or *C. pneumoniae*-infected neutrophils were treated with ER stress inhibitor 4μ8C or MitoTEMPO. mRNA of neutrophils was isolated at 12 h.p.i. and translated to cDNA. *Xbp1* was amplified by PCR and then digested with PstI enzyme for 2 h. The ratio of s*Xbp1* to total *Xbp1* was normalized to the control *C. pneumoniae*-infected neutrophils. (**I**,**J**) Mock or *C. pneumoniae*-infected neutrophils were treated with mtROS scavenger MitoTEMPO or IRE1α inhibitor 4μ8C. Neutrophils were collected at 12 h.p.i., stained with XBP-1S antibody, and then analyzed by flow cytometry. (**K**) Neutrophils infected with *C. pneumoniae* were treated with the indicated dose of MitoTEMPO. Culture medium was collected 24 hrs after infection. TNFα and IL-10 in the culture medium were measured by ELISA. (**L**,**M**) Neutrophils infected with *C. pneumoniae* were treated with ER stress inhibitor 4-PBA (L) or 4μ8C (**M**); culture medium was collected 24 h after infection. IL-10 in the culture medium was measured by ELISA. Statistical analyses were performed using a two-way ANOVA followed by Tukey’s multiple comparison test (**A**,**B**), an unpaired two-tailed Student’s *t*-test (**C**), and a one-way ANOVA followed by Tukey’s multiple comparison test (**F**,**J**–**M**). * *p* < 0.05, ** *p* < 0.01, *** *p* < 0.001, and **** *p* < 0.0001. Every experiment was conducted with n ≥ 3 and repeated at least three times, and the experiment shown is representative. Cpn: *C. pneumoniae*.

**Figure 3 cells-14-01523-f003:**
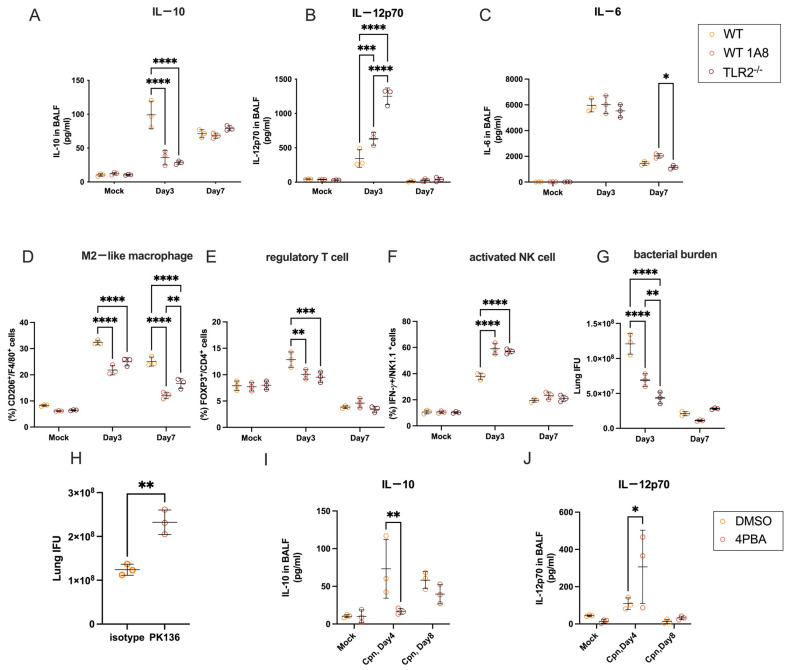
In vivo, neutrophil-derived IL-10 shapes immune responses in the lungs of mice infected with *C. pneumoniae.* (**A**–**C**) C57BL/6J (WT) mice, WT mice treated with 1A8 mAb, or C57BL/6J TLR2^−/−^ mice were intranasally infected with *C. pneumoniae*. Bronchoalveolar lavage fluid (BALF) was collected from infected mice at the indicated days after infection. IL-10 (**A**), IL-12p70 (**B**), and IL-6 (**C**) in the BALF were measured by ELISA. (**D**–**F**) WT mice, 1A8 mAb-treated WT mice, or TLR2^−/−^ mice were nasally infected with *C. pneumoniae*. Infiltrated leukocytes in the lungs were isolated at the indicated days after infection and analyzed by flow cytometry. MFI of CD206 on macrophages (**D**), percentage of CD4^+^Foxp3^+^ cells within CD4^+^ cells (**E**), and percentage of NK1.1^+^IFNγ^+^ cells within NK1.1^+^ cells (**F**). (**G**) C57BL/6J mice, C57BL/6J mice treated with 1A8 abs, or TLR2^−/−^ mice were nasally infected with *C. pneumoniae*. Lungs were collected on the indicated days after infection. The infectious burden of *C. pneumoniae* in the lungs was measured by IFU assay. (**H**) C57BL/6J mice or PK136 abs-treated C57BL/6J mice were intranasally infected with *C. pneumoniae.* IFU of *C. pneumoniae* in the lungs was measured. (**I**,**J**) C57BL/6J mice were nasally infected with *C. pneumoniae* and treated with solvent or 4-PBA. BALF was collected from infected mice on the indicated days after infection. IL-10 (**I**) and IL-12p70 (**J**) in the BALF were measured by ELISA. Statistical analyses were performed using a two-way ANOVA followed by Tukey’s multiple comparison test (**A**–**G**,**I**,**J**) and an unpaired two-tailed Student’s *t*-test (**H**). * *p* < 0.05, ** *p* < 0.01, *** *p* < 0.001, and **** *p* < 0.0001. Every experiment was performed with at least three mice and repeated at least three times, and the experiment shown is representative. Cpn: *C. pneumoniae*.

**Figure 4 cells-14-01523-f004:**
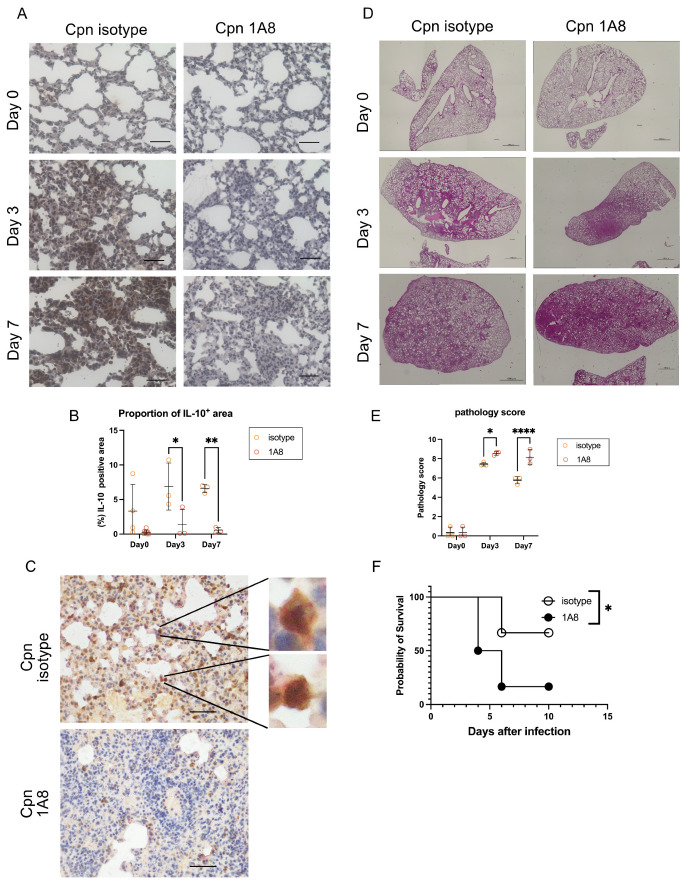
In vivo, neutrophil-derived IL-10 reduces lung tissue damage during *C. pneumoniae* lung infection. (**A**–**G**) Mice treated with isotype-matched Abs or Ly6G-specific 1A8 Abs were intranasally infected with *C. pneumoniae*. (**A**) Lung tissues on days 0, 3, and 7 after infection were stained with anti-IL-10 mAb (brown). The scale bar indicates 50 µm. (**B**) The proportion of IL-10-positive staining area was analyzed using ImageJ software. (**C**) Lung tissues on day 3 post-infection were double-stained with anti-IL-10 (brown) and chloroacetate esterase (red) for identifying IL-10-producing neutrophils. The scale bar indicates 50 µm. (**D**,**E**) H&E staining of the lung tissues on days 3 and 7 after infection and the pathological scores of the lungs from mice treated with isotype-matched control Abs or 1A8 Abs after *C. pneumoniae* lung infection. The scale bar indicates 1000 µm. (**F**) Mice treated with isotype Abs or 1A8 Abs were then infected with *C. pneumoniae*. Survival (n = 6) was measured at the indicated days. Statistical analyses were performed using a two-way ANOVA followed by Tukey’s multiple comparison test (**B**,**E**) and the log-rank test (**F**). * *p* < 0.05, ** *p* < 0.01, and **** *p* < 0.0001. Experiments (**F**) were repeated three times, and the presented experiment is representative. Cpn: *C. pneumoniae*.

**Figure 5 cells-14-01523-f005:**
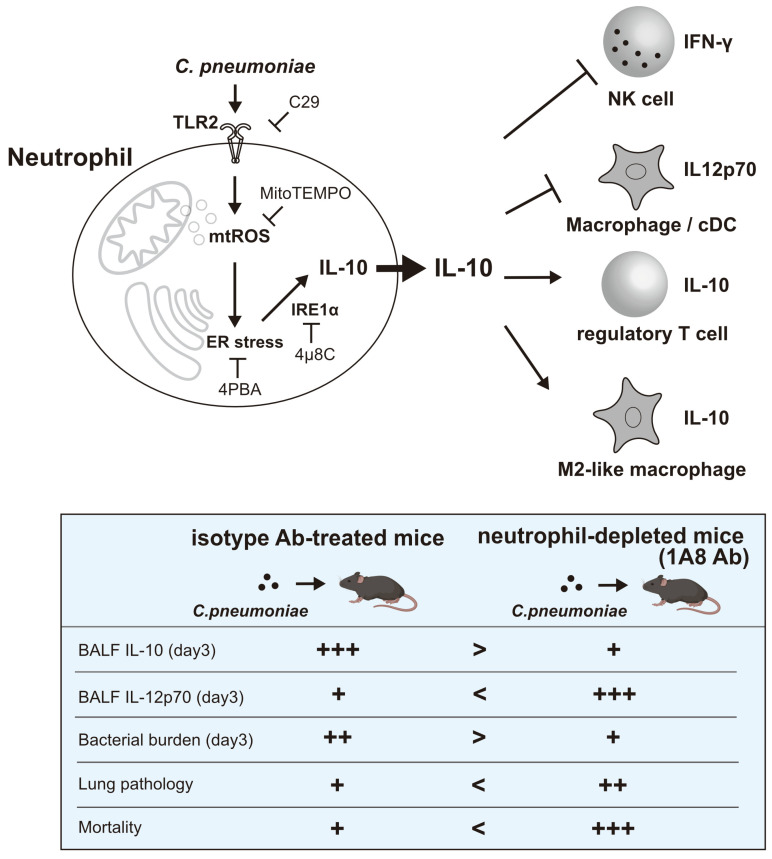
Induction of IL-10 in neutrophils and their roles in lung infection with *C. pneumoniae*. The induction of IL-10 in *C. pneumoniae*-infected neutrophils depends on the activation of the TLR2–mtROS–ER stress axis. IL-10 produced by neutrophils early in infection suppresses IL-12 p70 production from macrophages and/or conventional DCs and inhibits Natural Killer (NK) cell activation, while promoting regulatory T cells and M2-like macrophages. These IL-10-producing neutrophils are crucial for maintaining immune balance during *C. pneumoniae* infection in the lungs. Blocking *C. pneumoniae* infection-induced IL-10 production from neutrophils through neutrophil depletion (1A8 Ab treatment) or in TLR2^−/−^ mice enhances immune responses against *C. pneumoniae* and reduces bacterial burden in the lungs. However, this heightened immune response can also cause excessive inflammation and tissue damage, leading to severe mortality. The symbols +, ++, and +++ indicate the relative degree (low, moderate, and high, respectively) of the various outcomes. Created in BioRender. Ozuru, R. (2025) https://BioRender.com/fzq2h4y.

## Data Availability

The original contributions presented in the study are included in the article; further inquiries can be directed to the corresponding author.

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
