# Peer review of "Mitochondrial ROS–ER Stress Axis Governs IL-10 Production in Neutrophils and Regulates Inflammation in Murine Chlamydia pneumoniae Lung Infection"

_cells, 2025, doi:10.3390/cells14191523_

Round 1

Reviewer 1 Report

Comments and Suggestions for Authors

This interesting paper provides convincing evidence to demonstrate a role of neutrophil-derived IL-10 in modulating lung inflammation and mouse surivival after Cpn infection. An underlying role for TLR2 signaling, mtROS and ER stress is convincingly demonstrated using multiple complementary approaches to block receptors and pathways or to deplete neutrophils alone or together with other cell populations. 

2 minor suggestions for improvement:

-Please specify which post-test was used along with One-way or two-way ANOVA for statistical analysis of each data set in the figure legends.

-'Interphase' on line 169 should be 'interface' as it refers to layers of a gradient and not a stage of mitosis.

Author Response

This interesting paper provides convincing evidence to demonstrate a role of neutrophil-derived IL-10 in modulating lung inflammation and mouse surivival after Cpn infection. An underlying role for TLR2 signaling, mtROS and ER stress is convincingly demonstrated using multiple complementary approaches to block receptors and pathways or to deplete neutrophils alone or together with other cell populations. 

Response: We thank the reviewer for his or her insightful and helpful comments on our manuscript, as they have significantly assisted us in improving the quality of our paper.

2 minor suggestions for improvement:

-Please specify which post-test was used along with One-way or two-way ANOVA for statistical analysis of each data set in the figure legends.

Response: We apologize for not previously including the statistical details in the original manuscript. As suggested, we have now clarified the post-test and whether a one-way or two-way ANOVA was used for each dataset in the figure legends.

-'Interphase' on line 169 should be 'interface' as it refers to layers of a gradient and not a stage of mitosis.

Response: We apologize for this typographical error in the original manuscript. We have corrected 'interphase' to 'interface' on line 169 as you pointed out.

Reviewer 2 Report

Comments and Suggestions for Authors

The authors describe that Neutrophils regulate not only pro-, but also anti-inflammatory immune responses during Chlamydia pneumoniae lung infection by producing IL-10. In C57BL/6 mice, TLR2 activation triggered mitochondrial ROS and ER stress (via IRE1α–Xbp1), inducing IL-10 secretion. Blocking this pathway or depleting neutrophils lowered IL-10, increased IL-12p70 and IFN-γ NK cells, reduced Tregs and M2 macrophages, and, while lowering bacterial load, caused severe lung damage, weight loss, and reduced survival. Thus, neutrophil-derived IL-10 maintained immune balance and prevented excessive inflammation. This finding is novel and interesting. The study is well conducted and the manuscript very nicely written. However, I need to raise some issues.

  1. The authors observe relatively early IL-10 induction, while others describe the IL-10 increase by far later (e.g. Jupelli et al, PLoS One 2013: day 28, buit not day 14 or 21). How do the authors interpret this discordance?
  2. The authors draw a picture in which the C. pneumoniae -induced IL-10 host response is almost exclusively mediated via TLR-2 recognition by PMN. This is somewhat surprising because monocytes and macrophages are known to mediate the IL-10 response upon bacterial infection, including C. pneumoniae. How do the authors explain this finding? Can they exclude an effect of 1A8 on monocytes and macrophages?
  3. The statistical analyses should be described in more detail either in the methods section or in the respective figure legend.
  4. Fig 2A and 2K: the results of the statistical analysis should be restricted to results that are meaningful for the study´s main message. How was the comparison performed? Multivariate?
  5. Fig 3: do the authors have BAL IL-10 data from sham infected mice?
  6. Fig 5: inflammation (2x typo)

Author Response

The authors describe that Neutrophils regulate not only pro-, but also anti-inflammatory immune responses during Chlamydia pneumoniae lung infection by producing IL-10. In C57BL/6 mice, TLR2 activation triggered mitochondrial ROS and ER stress (via IRE1α–Xbp1), inducing IL-10 secretion. Blocking this pathway or depleting neutrophils lowered IL-10, increased IL-12p70 and IFN-γ NK cells, reduced Tregs and M2 macrophages, and, while lowering bacterial load, caused severe lung damage, weight loss, and reduced survival. Thus, neutrophil-derived IL-10 maintained immune balance and prevented excessive inflammation. This finding is novel and interesting. The study is well conducted and the manuscript very nicely written. However, I need to raise some issues.

We appreciate the reviewer for his or her thoughtful and constructive feedback on our manuscript, as it has greatly helped us improve the quality of our paper.

  1. The authors observe relatively early IL-10 induction, while others describe the IL-10 increase by far later (e.g. Jupelli et al, PLoS One 2013: day 28, but not day 14 or 21). How do the authors interpret this discordance?

Response: We thank the reviewer for this important question regarding the kinetics of IL-10 production and its source. We reported early IL-10 production in the lung (bronchoalveolar lavage fluid), while Jupelli et al. reported a significant increase in IL-10 levels in lung lysates at day 28 post-infection. We believe this apparent discordance can be explained by the time-dependent changes in the immune response, which involves IL-10-producing neutrophils that appear early during infection (on day 3 post-infection), and IL-10-producing alternatively activated (M2) macrophages that emerge later, as reported by Jupelli et al. In this study, we focused on the early immune response and found that early-appearing IL-10-producing neutrophils play a vital role in influencing (directing) the course of acquired immunity. We observed that the proportion of M2-like macrophages, which can produce IL-10, decreased at day 3 and day 7 post-infection in the lungs of neutrophil-depleted or TLR2-deficient mice (Figure 3D). We believe these M2-like macrophages and regulatory T cells can be important sources of IL-10 during the later stage of infection. Another factor contributing to the apparent discordance (day 3 vs. day 28) is that the strain of C. pneumoniae used in our study differs from that in their study (AR-39 vs. CM-1). The differences in virulence used in the mouse lung infection study could influence the course of lung infection (severity, duration, acute vs. chronic, time course, etc.), and it would be interesting to examine whether high or low virulence impacts the appearance of IL-10-producing neutrophils during the early innate immune response and the subsequent M1/M2 polarization during the acquired immune response.

  1. The authors draw a picture in which the C. pneumoniae -induced IL-10 host response is almost exclusively mediated via TLR-2 recognition by PMN. This is somewhat surprising because monocytes and macrophages are known to mediate the IL-10 response upon bacterial infection, including C. pneumoniae. How do the authors explain this finding? Can they exclude an effect of 1A8 on monocytes and macrophages?

Response: We appreciate the reviewer’s valuable and clarifying suggestion regarding the interpretation of our findings, especially concerning the scope of Figure 5. In response to the concern about exclusivity, we have revised Figure 5 and added a clarifying table or note to explicitly state that the Neutrophil TLR2–mtROS–ER stress axis mechanism shown is active early during infection (Day 3). We do not claim that infection-induced IL-10 production is solely mediated by neutrophils. The effect of neutrophil-depletion on IL-10 levels in BALF was significant at Day 3 but not at Day 7. This indicates that IL-10-producing cells shift from innate neutrophils to other cell types, such as M2-like macrophages and regulatory T cells. Importantly, depleting IL-10-producing neutrophils affects the appearance of M2-like macrophages (Fig. 3D). Regarding the specificity of 1A8 (anti-Ly6G specific), we included flow cytometry data in Supplemental Figure S3, which demonstrates that 1A8 treatment selectively depletes neutrophils but not macrophages. We believe this conclusively confirms that key monocyte or macrophage subsets are not depleted, ensuring that the reduction in IL-10 in BALF at Day 3 post-infection results solely from neutrophil removal.

  1. The statistical analyses should be described in more detail either in the methods section or in the respective figure legend.

Response: We apologize for not providing enough detail about our statistical analyses in the original manuscript. As requested, we have now included the specific statistical methods used for each data set in the corresponding figure legends to enhance clarity.

  1. Fig 2A and 2K: the results of the statistical analysis should be restricted to results that are meaningful for the study´s main message. How was the comparison performed? Multivariate?

Response: We apologize for the confusion about the statistical analysis. The comparisons in Fig. 2A were done using two-way ANOVA, while those in Fig. 2K used one-way ANOVA. We have clarified this difference and specified the analysis method in the figure legends for both figures.

  1. Fg 3: do the authors have BAL IL-10 data from sham infected mice?

Response: Thank you for this important question. We apologize for the confusion caused by our use of 'Day 0', which was intended to represent mock-infected mice. We have now clarified this point and changed all instances of 'Day 0' to 'Mock' in Figure 3A-3F to more accurately represent the data from sham-infected mice.

  1. Fig 5: inflammation (2x typo)

Response: We apologize for the typographical error in Figure 5. We thank the reviewer for their careful reading, which helped us identify this mistake. In addition to fixing the spelling of 'inflammation,' we used this opportunity to revise the figure to better highlight the key findings of our study.

Round 2

Reviewer 2 Report

Comments and Suggestions for Authors

I´m grateful that the authors acted adequately on each and every suggestion. I have no further suggestions.